# Mutual Coupling Suppression of GPR Antennas by Depositing Wideband Meta-Absorber with Resistive Film

**DOI:** 10.3390/ma15207137

**Published:** 2022-10-13

**Authors:** Yajun Zhou, Minjie Guo, Linyan Guo, Yi Zhou, Changxin Wei

**Affiliations:** 1School of Geophysics and Information Technology, China University of Geosciences, Beijing 100083, China; 2College of Physics and Electric Engineering, Guangxi Normal University for Nationalities, Chongzuo 532200, China; 3School of Materials Science and Technology, China University of Geosciences, Beijing 100083, China

**Keywords:** mutual coupling suppression, wideband absorber, resistive film, Minkowski fractal, GPR antenna

## Abstract

The direct wave between the transceiver antenna negatively affects the dynamic range and imaging quality of ground penetrating radar (GPR). Suppressing this direct wave is a vital problem in enhancing the performance of the whole GPR system. In this paper, a Minkowski-fractal metamaterial absorber (MMA) with the resistive film is proposed in the GPR transceiver antenna to reduce the mutual coupling. The simulated and measured results indicate that this MMA has an effective wideband absorption in 1.0-8.0 GHz. And the thickness of MMA is only 0.007 $\lambda_{0}$ (with respect to 2.0 GHz). This wideband MMA can reduce the mutual coupling of the proposed GPR transceiver antenna by an average of 10 dB. And it also mitigates the time-domain ringing problem of the transmit antenna. Real-world experiments demonstrate that the direct wave from the transmitting antenna can be reduced and the target echo arriving at the receiving antenna can be increased if this MMA is placed in the proposed transceiver antenna. This resistive film-based MMA offers great promise in realizing low-cost, compact, and lightweight GPR antennas, which can also be extended to high-frequency microwave imaging.

## 1. Introduction

Ground Penetrating Radar (GPR), as non-destructive testing equipment in geophysical exploration, has always attracted much attention [1,2,3,4]. It transmits electromagnetic waves directionally utilizing the transmitting antenna, processes and analyzes the echo by the receiving antenna, and locates the distribution of the target objects according to the electrical difference of the underground media or other target objects [5,6]. It has been widely used and developed due to its high sensitivity, low hardware performance requirements, and high electromagnetic susceptibility [7]. The performance of the antenna will affect the efficiency and accuracy of the entire GPR system.

However, there are serious signal interference problems between the transceiver antennas. It is mainly due to the strong direct wave interference between the transceiver antennas and the strong reflection signal from the ground surface. The strength of the mutual coupling effect between the transceiver antennas is several times higher than that of the echo, which reduces the sensitivity of the receiver and causes the receiver front-end to be saturated or even damaged. Many schemes have been proposed for how to suppress the coupling interference between the transceiver antennas [8,9,10,11,12]. The conventional method is to adjust the distance between the transceiver antenna to reduce the mutual coupling [8]. Normally, it needs to exceed 0.5 $\lambda_{0}$ (with respect to the center frequency) to ensure the interference is low enough, which causes an overlarge volume of the GPR antenna [9]. Therefore, the other common and effective method to reduce mutual coupling is using some medium blocking between the transceiver antenna, such as absorber material [10,11] and transceiver separated back-cavity [12].

This method of using absorbing materials to block the transceiver antenna and laying them in the metal back-cavity at the same time is common in commercial GPR antennas. It helps to enhance the transceiver antenna isolation and reduce the time-domain ringing effect with no inference to the radiation performance. However, the traditional commercial absorbing materials are expensive, high profile, and high density. Against these problems, some research starts to use the low-profile metamaterial absorber to decouple in GPR transceiver antenna [13]. Because of the limited bandwidth in those traditional metamaterials, some recent studies expand the bandwidth by the combination method [14,15,16], loading lumped elements [17,18,19,20,21,22], using resistive film layer [23,24,25], and so on. But they usually work at high frequency and have little application in GPR antennas.

Inspired by those metamaterial absorbers in high frequency, this paper introduces a metamaterial absorber based on the resistive film, which is applicated in a GPR transceiver antenna to reduce the mutual coupling. This design uses the Minkowski-fractal metamaterial absorber (MMA) to reduce the interference of direct waves and mitigate the time-domain ringing problem between transceiver antenna while maintaining a miniaturized structure. The rest of this paper is organized as follows. Section 2 presents the design of the GPR transceiver antenna and its electromagnetic performance. The analysis of the metamaterial absorber is discussed in Section 3. The fabrication and experimental results are presented in Section 4. And the conclusion is given in Section 5.

## 2. Design and Analysis of the GPR Transceiver Antenna

Figure 1 illustrates several unavoidable interferences between the transceiver antenna in the GPR system. They affect the radiation performance of the antennas. And the direct wave is one of the strongest interferences, which causes the severe mutual coupling between the transceiver antenna. Besides, the strong electromagnetic interference between antennas and back-cavity causes the induced current, which raises the time-domain ringing problem.

During these situations, this paper designs a metamaterial absorber for decoupling and mitigating signal tailing. To reduce the mutual coupling and optimize the radiation performance of the antennas, a double-sided MMA is added between the two bow-tie antennas. At the same time, the bottom of the back-cavity uses MMA instead of traditional absorbing material to reduce the depth of the back-cavity and limit the radiation tailing. The specific structure and radiation performance are presented as follows.

### 2.1. The Transceiver Antenna Configuration

The overall GPR transceiver antenna is shown in Figure 2a. MMA is supplied in the middle of the transceiver antenna and inside the back cavity. The outer dimension of the back-cavity antenna remains the same with and without MMA.

This MMA unit is shown in Figure 2b. It is designed with a three-layer structure: the top metal geometry, the middle dielectric layer, and the underlying backplane. The top metal geometry is fabricated by a copper film with a thickness of $h_{1}=0.03$ mm. Its electrical conductivity is $5.8\times 10^{7}$ S/m. The dielectric layer is made of FR-4 with a thickness of $h_{2}=1$ mm. Its relative dielectric constant is 4.3, and the loss tangent is 0.025. The backplane is made of special resistive film material with an equivalent sheet resistance R=700 kΩ with a thickness of $h_{3}=0.0254$ mm. The specific parameters of the geometric structure are $S_{1}=4.3$ mm, $S_{2}=7.4$ mm, $S_{3}=6.3$ mm, $S_{4}=8.3$ mm, and the line width $W_{t}=0.6$ mm. And the size of a unit cell is P=40 mm. This MMA is arranged the transceiver antenna middle and the back-cavity inside.

Using the antenna design of the previous research [26], the bow-tie antenna uses an FR-4 dielectric plate with a thickness of $h_{4}=1.5$ mm to separate the metal patches on the top and the bottom layer. The metal patches on the top and the bottom layer and the feed lines are etched to copper with a thickness of $h_{5}=0.03$ mm. The specific antenna structure and the detail parameters are shown in Figure 2c, where $L_{1}\;=\;78$ mm, $L_{2}=55$ mm, $L_{3}=5$ mm, $L_{4}=15$ mm, $L_{5}=32$ mm, $W_{1}=110$ mm, $W_{2}=39$ mm, $W_{3}=39$ mm, $W_{4}=2.6$ mm, $W_{5}=4$ mm.

Considering the size limitation of the low-frequency antenna, the distance between the two bow-tie antennas is set to D = 10 mm (0.067 $\lambda_{0}$ with respect to the center frequency). According to the high directivity requirement of the GPR antenna, a metal back-cavity is designed to improve the antenna’s emission gain and directivity, and also to suppress the backward radiation since the bow-tie antenna is a bidirectional radiating antenna. The structure of the transceiver antenna loaded with the metal back-cavity is shown in Figure 2d,e. The distances from the transceiver antenna to the edges of the back-cavity are $D_{1}=60$ mm and $D_{2}=80$ mm, respectively. The back-cavity is designed with the copper with a thickness of $H_{t}=1$ mm, and the outer dimensions are L=352 mm, W=240 mm, and H=46 mm.

### 2.2. The Transceiver Antenna Performance

The proposed GPR transceiver antenna is simulated using CST Microwave Studio, and the simulated results are presented in Figure 3. The reflection coefficient ($\left|S_{11}\right|$) in Figure 3a is maintained below −10 dB at 1.0–3.0 GHz, and there is minor difference whether the model with or without MMA. This indicates that loading MMA has no obvious influence on the reflection performance of the antenna structure. However, considering the overall size limitation of the antenna structure, the distance between the two antennas needs to be as small as possible, and there is a certain mutual interference between the transmission and receiver antennas. As Figure 3b shows, before loading MMA, the average transmission coefficient ($\left|S_{21}\right|$) is about −15 dB, and the mutual coupling phenomenon is relatively obvious. The $\left|S_{21}\right|$ is significantly reduced after loading MMA, and the average reduction is about 10 dB. At the strong coupling frequencies, the reduction range can reach 15 dB, such as 1.0 GHz and 2.8 GHz. This shows that MMA can achieve the effect of decoupling and improving the isolation between the transceiver antenna indeed.

Meanwhile, the time-domain tailing situation of the antenna would worsen by loading the back-cavity. This problem is also considered to be solved by loading MMA. In Figure 3c, Marker 1 shows the time-domain signal disappears at 4.54 ns after loading MMA at the bottom of the back-cavity and in the middle of the transceiver antenna. And the result without MMA is held after 8 ns. Meanwhile, the normalized amplitude of the time-domain signal is suppressed by 20%, which indicates the signal ringing problem is alleviated.

Besides, it also can analyze the mutual coupling situation through the envelope correlation coefficient (ECC) [27]. Low ECC indicates good independence and diversity gain between antennas. Combining the S parameter and the radiation efficiency, and ECC is then given by
(1)$$\rho_{e,ij}=\frac{{\left|S_{ii}^{*}S_{ij}+S_{ji}^{*}S_{jj}\right|}^{2}}{\left(1-{\left|S_{ii}\right|}^{2}-{\left|S_{ji}\right|}^{2}\right)\left(1-{\left|S_{jj}\right|}^{2}-{\left|S_{ij}\right|}^{2}\right)\eta_{i}\eta_{j}}$$
where $\eta_{i}$ and $\eta_{j}$ are the radiation efficiencies of antenna i and j, respectively. The calculated ECC of the antenna system with and without MMA is shown in Figure 3d. The ECC of the transceiver antenna is reduced after loading MMA, which indicates that it is an effective approach to improve the performance of GPR transceiver antenna systems.

The simulated radiation patterns of the transceiver antenna with and without MMA at 1.0 GHz, 2.0 GHz, and 3.0 GHz are shown in Figure 4. By analyzing and comparing the simulated results, there is no obvious difference between these radiation patterns. It indicates that loading MMA would not deteriorate the radiation performance of the transceiver antenna.

The simulated gain comparison of the proposed antenna is shown in Figure 5a. The maximum gain of the proposed transceiver antenna without MMA can reach 9.97 dB. After loading the decoupled MMA, the gain of the transceiver antenna will be reduced. But compare to the transceiver antenna without back-cavity, it still increased by up to 2.8 dB. The maximum gain of the proposed transceiver antenna with MMA can reach 6.46 dB. The transceiver antenna with MMA has a 3 dB gain bandwidth from 1.3 to 3.2 GHz (the percentage bandwidth is 90.4%). And the simulated front-to-back ratio maintains well between 10 dB to 15 dB, as shown in Figure 5b. As with the gain, the radiation efficiency drops from 0.85 to 0.57, losing almost 50%. Although loading MMA decreases the antenna gain and radiation efficiency, this research mainly focus on the low-cost metamaterial absorber, the mutual coupling suppression, and the signal ringing problem alleviation. And it still meets the standard of GPR antenna gain.

Combined with the above results, it can be concluded that loading MMA not only can decrease the obvious coupling problem between the transceiver antenna but also can mitigate the signal ringing in the time domain. Meanwhile, the bandwidth and the directivity are still maintained basically the same as the transceiver antenna without MMA. Thus, loading MMA would not deteriorate the radiation characteristics of the transceiver antenna.

## 3. The Absorption Mechanism of the MMA

The absorption mechanism of the MMA is examined in-depth by simulating five comparative cases shown in Figure 6. The Minkowski fractal loop structure electric resonator is shown in Figure 6a. Case I is the Minkowski fractal loop resonator on top of an FR-4 dielectric plane without the metal backplane, as Figure 6b shows. In Figure 6c, Case II covers a 0.03 mm-thick copper film in the bottom layer as the conventional metamaterial absorber. And in Figure 6d, Case III is the proposed MMA unit cell. It uses a 0.0254 mm-thick resistive film with an equivalent sheet resistance R=700 kΩ. Case IV is the same as Case III except without the fractual structure and Case V is only the resistive film. In all five cases, the same material layers have the same thickness. So these factors would not have an impact on the simulated results.

The S parameter of materials can be used to calculate their basic electromagnetic properties, such as material absorptance [28]. And the material absorptance is given by
(2)$$A\left(\omega \right)=1-{\left|S_{11}\right|}^{2}-{\left|S_{21}\right|}^{2}$$

According to the full wave simulated results using CST Microwave Studio, the absorptance of the MMA can be obtained. As Figure 7 shows, Case I has three main absorptance peaks at 5.5 GHz, 6.7 GHz, and 7.6 GHz because of the resonance of the Minkowski fractal structure. And the absorptance peaks of Case II are at 1.8 GHz, 3.0 GHz, 4.2 GHz, 5.3 GHz, and 6.4 GHz, respectively. However, the absorptance of Case III is always maintained above 85% at 1.0–8.0 GHz, with no significant absorptance peaks. The absorptance of Case IV is similar to Case V. They are over 60% at 1.0–8.0 GHz and decrease with increasing frequency.

And then, the simulation results analyze the power loss in different mediums to figure out the difference in the absorption mechanism of the five cases. As shown in Figure 7b,c, it depicts the normalized amplitude of power loss in two different mediums. The power loss in FR-4 basically fits the absorptance peaks in Case I and Case II, which means the main absorption in Case I and Case II comes from the structure resonance. But for Case III, it is a really low power loss compared to the absorption. Combined with the simulated results in conducting medium in Figure 7c, it shows the power loss in Case III is all dependent on the conducting medium, which is nearly zero in Case I and Case II. It indicates the resistive film provide strong power loss in the MMA.

To validate the above conclusion, the power loss density distributions of three cases are examined at different frequencies, as shown in Figure 8. And three typical frequencies are also pointed out in Figure 7b for analysis. The cut plane is at the half-thick of the dielectric layer in each case. For Case I and Case II, the power loss in the red color area, which indicates a maximum power loss, are mainly concentrated in the central area of the Minkowski fractal structure at resonance frequencies (e.g., typically, 1.8 GHz and 3.0 GHz). At non-resonance frequency (e.g., 2.2 GHz), the power loss of Case II is almost in the blue color area, which means a minimum loss. The power loss density distributions of three cases can again confirm the simulated results in Figure 7b. Conversely, for Case III, the power loss is mainly focused on the outside area of the Minkowski fractal structure (see the green color area). But there still have some power loss in the resonance structure (see in the rad color area). Despite that, the area of the power loss density distributions of Case III is wider than it is in Case I and Case II. Therefore, based on the intriguing power loss feature shown in Figure 8, it can deduce that the wideband high absorption of the MMA originates mainly from both the resonance property of the Minkowski fractal structure and the Ohmic loss property of the resistive film.

The angular stability analysis of the metasurface is critical in the application [23,29,30,31]. According to this, the wide-angle property is investigated at the simulated absorptance in different incident angles as shown in Figure 9. The absorptance has no significant influence when the incident angle is below 10°. And the absorptance maintains over 85% in 1.0–8.0 GHz when the incident angle is from 10° to 30°. When the incident angle is 40°, the absorptance drops to about 80%. The absorptance is close to 75% in 50°. These results indicate this MMA unit cell design has stable absorption performance when the incident angle is below 40°.

Combined with the above results, the Minkowski fractal structure can produce high power loss at specific resonance frequencies in all three cases. Compared with the conventional metamaterial absorption mechanism, the resistive film can provide the wide-area power loss through the Ohmic loss property, which can widen the frequency band. And in the oblique incident condition, this MMA unit cell design shows stable absorption performance when the incident angle is under 40°.

## 4. Antenna Fabrication and Measurement Results

Based on the above simulated results, the experiments are conducted to confirm their reliability. The entire bow-tie antenna and the part of the MMA are just fabricated using printed circuit board (PCB) technology, which is already industrialized.

Compared with the conventional metamaterial absorbers, the most special part is to manufacture resistive film with a specific square resistance value. The basic material of the resistive film is carbon conductive paste, also known as conductive ink. It is a stable commercial material and common applicate in medical electronics, membrane switches, and communication equipment. The actual achievable square resistance range is wide, spanning multiple orders of magnitude. The main components of this carbon conductive paste chosen in this paper are shown in Table 1. The sample is placed on the platform and secured to ensure it would not move in the coating. The sepecific wet film coater depends on the size of the sample and the required resistive film thickness. The coater should be close to the edge of the sample. About 5 mL of the carbon paste is taken in front of the preparation device each time, and then the device is pushed at 150 mm per second to apply a wet film of desired thickness. After that, the sample stands horizontally until the wet film is completely dry. Due to the organic solvent in the conductive carbon paste, the process of coating must be carried out in a fume hood. At the same time, operators need to wear goggles and masks to prevent accidental contact of the mouth and nose and volatilization of substances. Conductive carbon paste is a black semi-solid with high viscosity. its viscosity is about 15,000–20,000 cp. Therefore, it can be firmly attached to the dielectric plate after it is completely dry. It would not come off easily with handling or friction. Therefore, the fabricated MMA has stable properties and is not easily disturbed by environmental factors. Considering those coating technologies and experimental situations, wet film preparation and silk-screen printing are usually used. And the appropriate screen plate can be customized so as to easily make the resistive film with a specific shape. As shown in Figure 10, the entire fabrication process about MMA and the proposed GPR antenna as the above described are exhibited.

The proposed transceiver antenna with the MMA is fabricated and tested in free space. The experiment results are as shown in Figure 11 and Figure 12. It can be seen that the overall trend of the measured results is consistent with the simulated results. The measured absorptance of MMA is shown in Figure 11a. Because of the influence from the fabrication process, the absorptance shows a few differences with the simulated result. But it still reaches 85%. And the measured S parameter of the proposed antenna with and without MMA are shown in Figure 11b,c. Figure 11b shows that the proposed GPR antenna with and without MMA works stably at 1.0–3.0 GHz. The tiny difference between the two results confirms that loading MMA does not affect the radiation performance of the proposed GPR transceiver antenna. Figure 11c shows that the mutual coupling is below −20 dB after loading MMA, which is 5–18 dB lower than before.

And the radiation properties of the proposed transceiver antenna with and without MMA at 1.0 GHz, 2.0 GHz, and 3.0 GHz are shown in Figure 12. In these frequencies, both the E-plane and H-plane radiation patterns are illustrated a forward directivity and a low back lobe. On the other hand, there is no obvious difference between these results, which can confirm that loading MMA have no influence to the radiation performance of the proposed GPR antenna.

An additional GPR experiment is conducted to confirm the availability of MMA in mutual coupling suppression, as shown in Figure 10f. The proposed transceiver antenna is secured on a trolley to ensure the radiation direction of this antenna is straight ahead. The survey line is perpendicular to the radiation direction. A metallic box is set as the target in the radiation direction. The front edge of this box is horizontal to the survey line. The distance between the target and survey line is 3 m. Figure 13 displays the GPR B-Scan images of the proposed transceiver antenna with and without the MMA. The measured results show the proposed antenna with MMA can depict a clearer B-Scan picture than without MMA. Apparently, the amplitude of the direct wave (in red squares) decreased while the reflection wave (in red circles) increased after placing MMA, which indicates that MMA can enhance the GPR antenna performance.

## 5. Conclusions

This paper fabricates a Minkowski fractal-based wideband MMA with resistive film, which is used in a GPR transceiver antenna to suppress the mutual coupling between the transceiver antenna and mitigate the time-domain ringing problem from the antennas and back-cavity. The simulated and measured results indicate that the MMA has above 85% absorptance at 1.0–8.0 GHz and the frequency band of this proposed transceiver antenna is 1.0–3.0 GHz. Through loading the MMA, the mutual coupling of the transceiver antenna is reduced by nearly 10 dB. Meanwhile, the radiation performance and bandwidth of the GPR antenna are maintained. Compared with traditional absorbers in commercial GPR shielding antenna, the proposed MMA has a light and thin structure with the properties of low cost and simple fabrication. The thickness of the MMA is only 1.054 mm, which is about 0.007 $\lambda_{0}$ (with respect to 2.0 GHz). The relative bandwidth of the transceiver antenna with MMA reaches 100%, which is suitable for pulse and step frequency GPR systems. The overall size of the antenna structure is small and low profile, which further realizes the miniaturization of the GPR system. By the application of the metamaterial absorbers with resistive film, the GPR transceiver antenna in low frequency could become more miniaturization and lightweight in the future.

## Figures and Tables

**Figure 1 materials-15-07137-f001:**
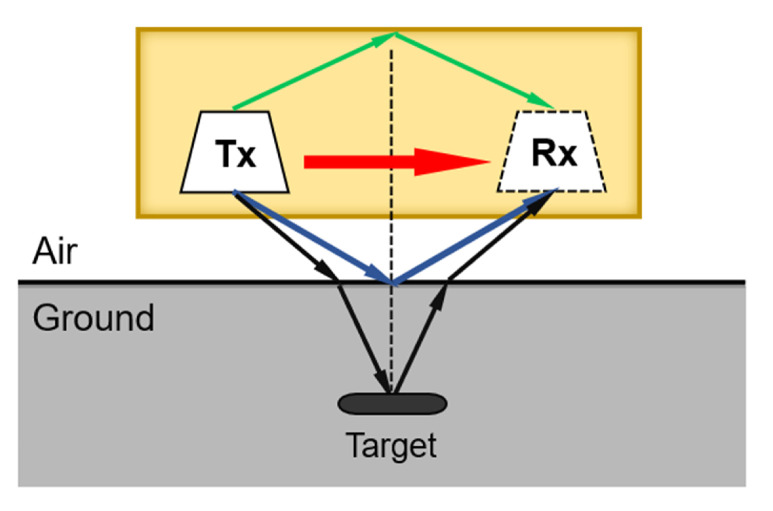
The prototype of the GPR transceiver antenna.

**Figure 2 materials-15-07137-f002:**
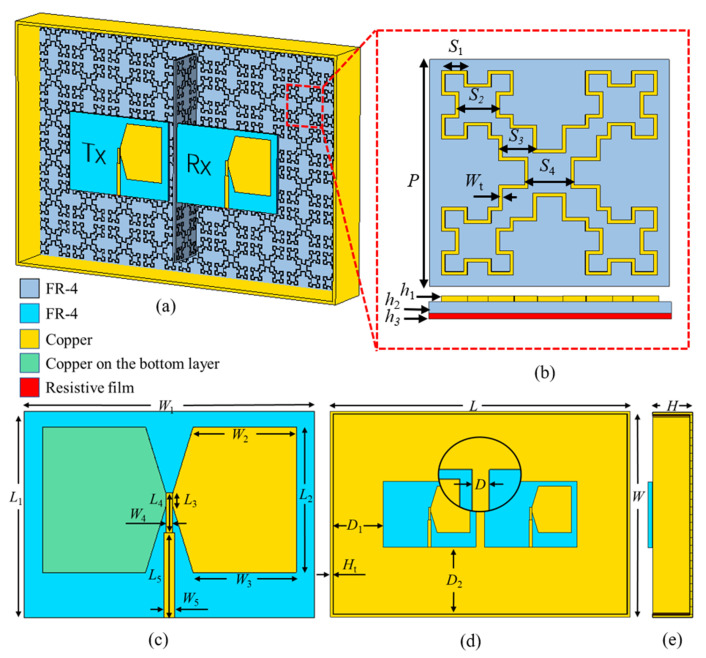
The proposed GPR transceiver antennas. (**a**) the overall structure. (**b**) the MMA unit. (**c**) the design of the bow-tie antenna. (**d**) the top view of the transceiver antennas with back-cavity. (**e**) the side view of the transceiver antennas with back-cavity.

**Figure 3 materials-15-07137-f003:**
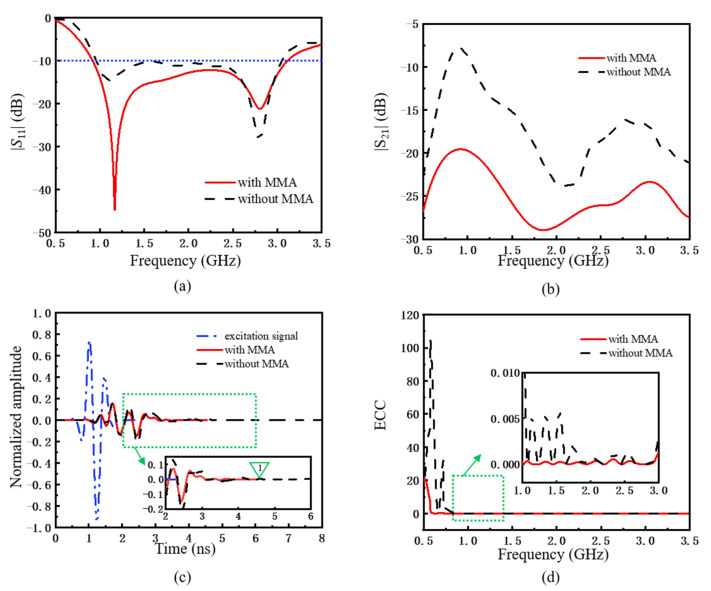
(**a**) $\left|S_{11}\right|$ and (**b**) $\left|S_{21}\right|$ of the transceiver antenna with and without MMA. (**c**) Time domain response of the transceiver antenna. (**d**) The ECC of the transceiver antenna.

**Figure 4 materials-15-07137-f004:**
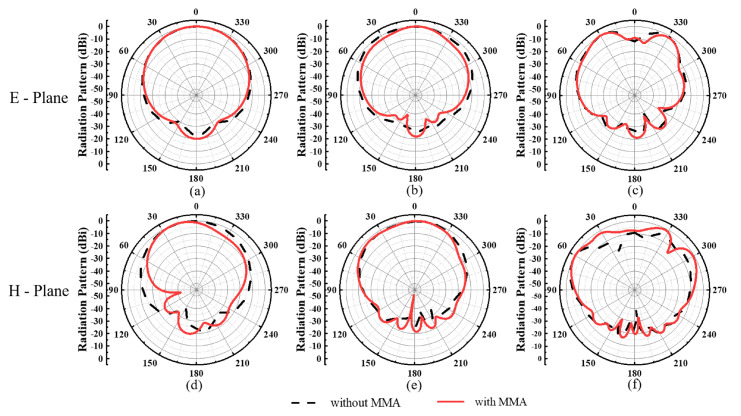
Simulated radiation patterns of the normalized amplitude of the transceiver antenna with and without the MMA. (**a**–**c**) E-plane and (**d**–**f**) H-plane at 1.0 GHz, 2.0 GHz,3.0 GHz, respectively.

**Figure 5 materials-15-07137-f005:**
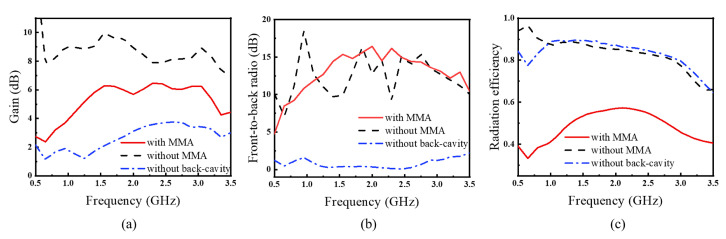
The simulated (**a**) gain, (**b**) front-to-back ratio and (**c**) radiation efficiency of the transceiver antenna.

**Figure 6 materials-15-07137-f006:**
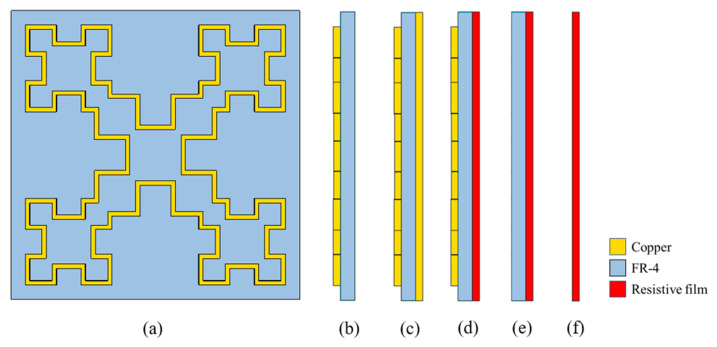
The schematic of (**a**) the Minkowski fractal structure. (**b**) Case I: without the back plane. (**c**) Case II: with the copper film. (**d**) Case III: the proposed MMA unit cell with resistive film. (**e**) Case IV: without the fractual structure. (**f**) Case V: only the resistive film.

**Figure 7 materials-15-07137-f007:**
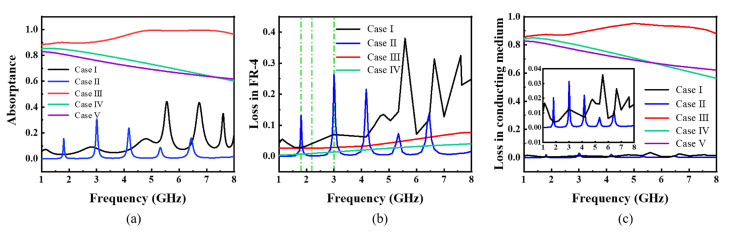
(**a**) The absorptance of five cases. Power loss in material. (**b**) in FR-4. (**c**) in conducting medium.

**Figure 8 materials-15-07137-f008:**
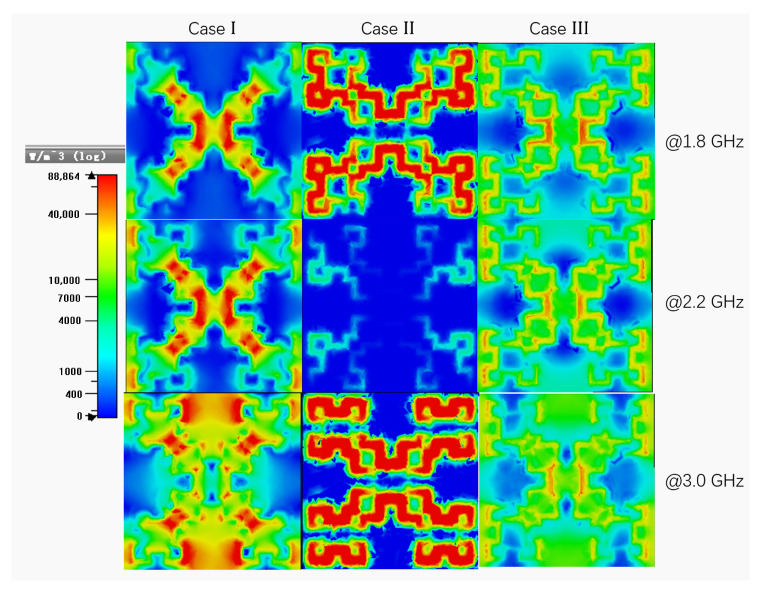
The power loss density distribution of three cases in FR-4 at xoy plane.

**Figure 9 materials-15-07137-f009:**
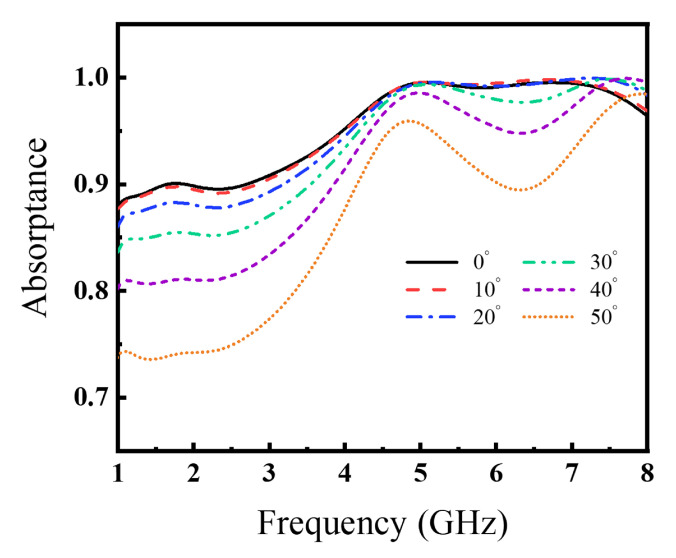
The absorptance at different incident angles.

**Figure 10 materials-15-07137-f010:**
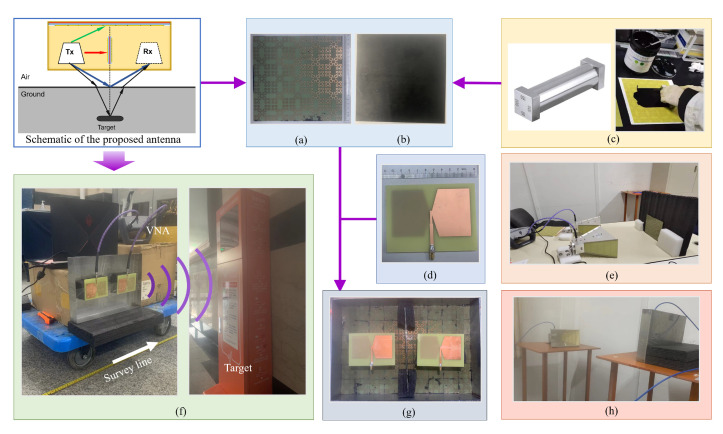
The photograph of the experiment equipment and the transceiver antenna. (**a**) The top view and (**b**) The back view of MMA. (**c**) The wet film coater (in left) and the resistive film process (in right). (**d**) The bow-tie antenna. (**e**) The absorptance measurement. (**f**) The setup of the reflection measurement (in left) and the metallic box (in right). (**g**) The proposed transceiver antenna with MMA. (**h**) The radiation parameter measurement.

**Figure 11 materials-15-07137-f011:**
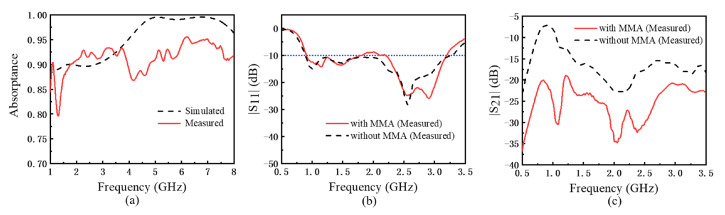
(**a**) The measured absorptance of MMA. The measured (**b**) $\left|S_{11}\right|$ and (**c**) $\left|S_{21}\right|$ of the proposed antenna with MMA and without MMA.

**Figure 12 materials-15-07137-f012:**
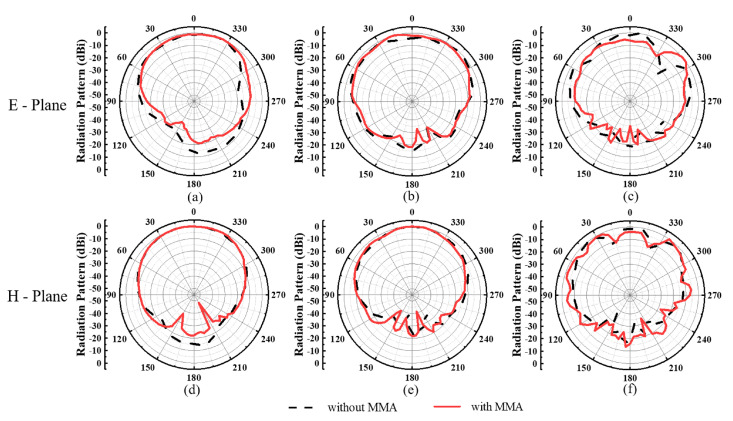
Measured radiation patterns of the normalized amplitude of the transceiver antenna with and without the MMA: (**a**–**c**) E-plane and (**d**–**f**) H-plane at 1.0 GHz, 2.0 GHz, 3.0 GHz, respectively.

**Figure 13 materials-15-07137-f013:**
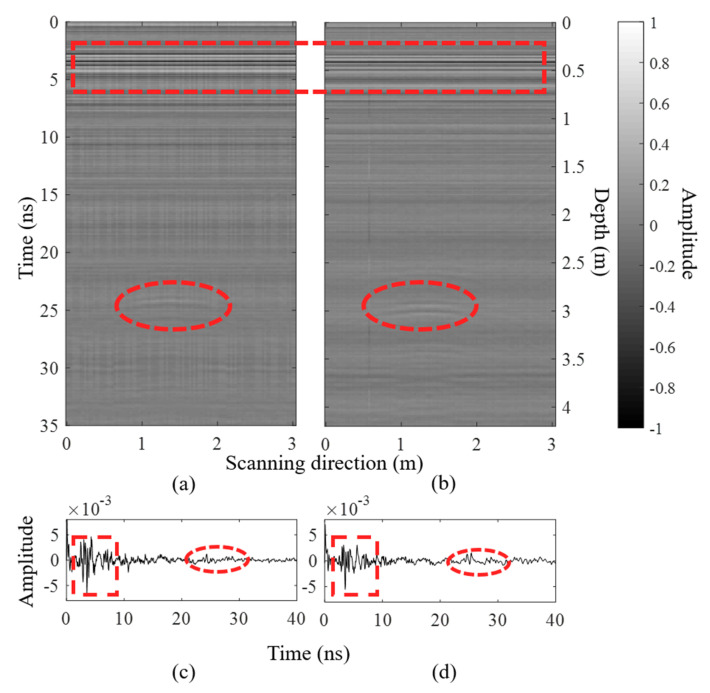
GPR B-Scan images for the transceiver antenna (**a**) without the MMA and (**b**) with the MMA. The return signal for the transceiver antenna (**c**) without the MMA and (**d**) with the MMA.

**Table 1 materials-15-07137-t001:** The main components of the carbon conductive paste.

Composition on Ingredients	Concentration (%)	CAS NO.
Isophorone	5%	78-59-1
Dibasic ester (DBE)	15%	95481-62-2
Carbon black	35–40%	341-11-11
Polyurethane	40%	51852-81-4

## Data Availability

Not applicable.

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
