# Peer review of "Mutual Coupling Suppression of GPR Antennas by Depositing Wideband Meta-Absorber with Resistive Film"

_materials, 2022, doi:10.3390/ma15207137_

Round 1

Reviewer 1 Report

The idea of using metasurfaces as absorbers in radio-frequency devices is a great way to reduce the dimensions. This work takes a good approach in utilizing metasurfaces to tackle technologically relevant issues.

Overall the article is well structured and coherent, and the first parts of the text are well written. The second half of the text though has some major language issues that eventually make it slightly difficult to understand some thoughts. The methodology in general is sound and results are quite comprehensive. Some questions still arise and some modifications would be appreciated.

Major comments:

1. Section 3 has some incomplete results. Fig.8 with accompanying discussion in the texts suggests that for the case with resistive back-film, most of the power loss is due to the resistive film. This raises the question, what would the performance be without the metamaterial fractal design on top. I think the comparison with another case should be made, which is Case III without the top copper metamaterial structure. If this additional case is not significant, these results can be put into supplemental material, but without it the results are not complete. Is the metastructure actually needed on top of the resistive film? Use of metallic back reflectors is a common practice and high Q-factors are expected, as the results in Fig.8 show. Broadening of spectral features due to increase in Ohmic losses is also well known.

2. In Section 4 it is not clear how the carbon paste film was deposited onto the samples in this work. This part needs a bit clarification.

3. The description of the last experiment with results in Fig.13 is not clear. The experiment description should be amended to make it clearer what was actually the experimental set-up.

4. On line 247 it is said that "Compared with traditional metamaterial absorbers". But in fact, light and thin structure are common traits of metasurfaces in general, not specifically this design. Maybe compared to traditional approaches in general, but I would not go so far to say that this structure is much thinner than metasurface structures in general.

Minor comments:

1. Fig. 2 has a well presented color legend, but 2(c) includes green color not shown in the legend. The reasons for showing the back lobe in different color are clear, but could be improved.

2. Fig. 2, 4, 9, 11, 12 and 13 include texts which are really small compared to the other texts and at times need extreme zooming to read.

3. Paragraph 2.2, line 102 states that MMA has no significant influence on the reflection coefficient (Fig.3(a)). Though the overall conclusion still holds, the dip at around 1.2GHz is pretty obvious and significant.

4. Line 115 has the statement that time-domain signal in Fig.3(c) shows significant difference between with and without MMA. Maybe it is just the representation (use of dashed line), but the difference does not look very significant on that figure.

5. Line 122 states that ECC of the antenna structure is under 0.5. The next sentence concludes that MMA is effective to improve GPR performance. While there is evidence that supports the latter statement, then the placement of this sentence right after the discussion about ECC is maybe not perfect. Because looking at Fig.3(d), ECC is well below 0.5 even without MMA. Maybe it is just wording and sentence placement that made it seem a bit misplaced.

6. Line 145 states that MMA does not influence the radiation characteristics. I think this firm statement is not quite correct, because it does influence it to an extent, just that the influence is not significant.

7. At which distance from the metasurface are the plots taken in Fig.9?

8. The experimental results were collected in free space. Would the results have been much different if the S-parameters and radiation patterns were measured in an anechoic chamber?

9. The thickness of the MMA is compared to the wavelenght at 4.5 GHz, but the working range of the proposed antenna is 1-3 GHz. Why not take lambda from this range?

Author Response

Dear Reviewer 1,

We appreciate very much the constructive and valuable comments, which have helped us a lot to improve the quality of the manuscript (ID: materials-1936271). Based on the comments and suggestions, we have revised the manuscript carefully and thoroughly. The explanations and corrections to all comments were added in the revised manuscript (in blue fonts). Below are our item-by-item responses (in red fonts) to the comments (in black fonts). We sincerely hope this manuscript could be finally accepted by Materials.

Thanks for your attention. Look forward to your reply. Have a nice day.

Best wishes,

Linyan Guo

Reviewer 2 Report

This contribution is interesting in terms of effects of combination of transceiver antenna with absorbing metasurface to reduce coupling effects. Although this idea is not new at all, there are some novel issues as the use of the resistive film combined with the metamaterial unit cells which enhances the overall performance as it is quite properly introduced and analyzed. 

In my opinion the paper deserves publication. However there are some issues that I would like to ask the authors to include in the manuscript for further clarification and to provide information to readers which could be interested for example in using these ideas to wearable radar applications at higher frequencies:

The gain, radiation efficiency and front-to-back ratio of the antennas without and with the metasurface should be included. There are other approaches to combine radar antennas and metasurfaces with effects on the mutual coupling between the antennas, but the impact on the antenna radiation parameters is a key aspect and depending on the application it can be critical. (The reader should know if the coupling reduction is at the expense of radiation efficiency reduction since, for example).

The angular stability analysis of the metasurface is also relevant in term of oblique incidence for this kind of applications. Authors should provide performance limitations of the chosen metamaterial unit cell for the operating frequency band under oblique incidence, at least in simulation. There are recent relevant works on this including measurements, as the ones of H. Fernandez et al. that could be read and cited:

H. Fernandez Alvarez, et al. "Angular Stability of Metasurfaces: Challenges Regarding Reflectivity Measurements [Measurements Corner]," in IEEE Antennas and Propagation Magazine, vol. 58, no. 5, pp. 74-81, Oct. 2016, doi: 10.1109/MAP.2016.2594018.

H. F. Álvarez et al, "Paving the Way for Suitable Metasurfaces’ Measurements Under Oblique Incidence: Mono-/Bistatic and Near-/Far-Field Concerns," in IEEE Transactions on Instrumentation and Measurement, vol. 69, no. 4, pp. 1737-1744, April 2020, doi: 10.1109/TIM.2019.2913721.

Finally, the overall thickness increase due to the introduction of the absorbing structure should be also clearly indicated.

This reviewer encourages the authors to address these comments which could be of great interest for the readers.

Author Response

Dear Reviewer 2,

We appreciate very much the constructive and valuable comments, which have helped us a lot to improve the quality of the manuscript (ID: materials-1936271). Based on the comments and suggestions, we have revised the manuscript carefully and thoroughly. The explanations and corrections to all comments were added in the revised manuscript (in blue fonts). Below are our item-by-item responses (in red fonts) to the comments (in black fonts). We sincerely hope this manuscript could be finally accepted by Materials.

Thanks for your attention. Look forward to your reply. Have a nice day.

Best wishes,

Linyan Guo

Round 2

Reviewer 1 Report

Thank you for the corrections and additions. The text is much more clear now. I would advise another round of proofreading to correct minor language and typing errors.